# A Retrospective Study of Epidemiological Correlations of Food, Drug and Chemical Poisoning in Al-Baha, Western Saudi Arabia

**DOI:** 10.3390/healthcare11101398

**Published:** 2023-05-11

**Authors:** Saba Beigh, Ali Mahzari, Read A. Alharbi, Rahaf A. Al-Ghamdi, Hanan E. Alyahyawi, Hind A. Al-Zahrani, Saeedah Al-Jadani

**Affiliations:** 1Department of Public Health, Faculty of Applied Medical Sciences, Albaha University, Al-Baha 65431, Saudi Arabia; 2Department of Laboratory Medicine, Faculty of Applied Medical Sciences, Albaha University, Al-Baha 65431, Saudi Arabia; 3Basic Sciences, College of Applied of Medical Sciences, Albaha University, Al-Baha 65431, Saudi Arabia

**Keywords:** toxicity, adverse drug effects, foodborne infections, chemical adverse effects

## Abstract

Poisoning is a common and severe problem worldwide. Due to significant growth in the agricultural, chemical, and pharmaceutical industries over the past few decades, poisoning risks have increased with the use of food, chemicals, and medicines everywhere in the world, especially in Saudi Arabia. Advanced information on acute poisoning patterns is critical for the effective management of poisoning events. This study aimed to examine the characteristics of patients with various patterns of acute poisoning, caused by food, drugs, and chemicals, that were reported to the Department of Toxicology and Poison Center at King Fahad Hospital and the Poison Center in Al-Baha Province, Saudi Arabia. The study also examined the relationship between demographic characteristics, including age, toxin type, and geographical distribution, and poisonings in Baha Province. This retrospective cross-sectional analysis included 622 poisoning cases. The data were collected from 2019 to 2022 and it was found that out of 622 instances, 159 had food poisoning, with more men than females sick (53.5% male and 46.5% female), 377 had drug poisoning (54.1% males and 45.9% females), and 86 had chemical poisoning (74.4% males and 25.6% females). This study found that the most prevalent agents implicated in acute poisoning were medicines, particularly analgesics and antipsychotic drugs. Food poisoning was the second most common acute poisoning, affecting largely males followed by female patients. Finally, chemical poisoning involved acute poisoning, with most cases involving methanol and household items including the strongest bleaches (chlorines) (Clorox^®^, Oakland, CA, USA). Insecticides and pesticides were also secondary sources of chemical poisoning. Additional research revealed that the incidence of food, chemical, and drug poisoning was highest in children aged 1–15 years (food poisoning, *n* = 105, 66%; drug poisoning, *n* = 120, 31.8%); patients aged 11–20 years had the highest incidence of chemical poisoning (*n* = 41, 47.7%). Most poisoning incidents among youngsters are caused by easy access to drugs at home. Implementing strategies to enhance public awareness and limit children’s access to drugs would contribute considerably to decreasing the community’s burden of this problem. The findings of this study suggest that Al-Baha should improve its education regarding the rational and safe use of drugs and chemicals.

## 1. Introduction

Poisoning is a worldwide public health concern that is regarded as one of the most serious medical contingencies because its multitude varies from one location to another [1]. Poisoning is generally recognized as a health exacerbation caused by the admittance of a contaminant (microorganism or anomalous substance) at varying concentrations into the body, whether it be food, chemicals, or even drugs in different forms. According to the World Health Organization, human poisoning is a major clinical emergency around the world [2]. Many people are becoming reconciled with the occurrence of poisoning cases, unaware of the extent of the danger that it may cause, as it may result in death if not considered [3]. Poisoning events are estimated to cause more than one million illnesses each year, with the number differing, respectively, depending on intentional cases associated with approved or unapproved chemicals [2]. The clinical signs and repercussions of poisoning can vary among individuals, depending on an array of variables such as age, sex, environment, immune response, chronic disease, and psychological and social stress [4]. Owing to the significant expansion of the agricultural, chemical, and pharmaceutical industries over the past few decades, poisoning risks have increased worldwide, particularly in Saudi Arabia [5]. Food poisoning, also known as foodborne illness, is caused by the consumption of tainted or unhealthy foods. Whether the food is consumed immediately or purchased for later consumption, the outbreak of food poisoning is linked to catering forms.

Acute contamination occurs when food is contaminated by microorganisms that aggravate toxicosis. *Salmonella* and *E. coli* have been reported to be among the most important pathogens that have recently become more prevalent causing poisoning, particularly in children [5]. Foodborne illnesses engendered by food products, such as beef and chicken, seem to be more widespread in Saudi Arabia, with *Salmonella* contamination being the second most common clinical manifestation [6]. Following the recent spread of COVID 19, in addition to food poisoning cases, other poisoning incidents have increased whether it is with chemical compounds or by inadvertently consuming large amounts of medication prescribed by a physician, generally resulting in toxic effects. Medicines are responsible for a huge spate of fatal overdoses [7]. In developed countries, unintentional poisoning (due to drugs) is more prevalent than in developing nations, with women having the highest prevalence rate. Drug poisoning is the second most dangerous cause of poisoning and results in deaths worldwide. It has surpassed smoking as the leading cause of death in many countries, including Brazil [8]. Because several people are unaware of the dangers of these compounds, some may overlook the importance of limiting their use unless necessary. Furthermore, the reduction in the consideration of some medicines, such as painkillers, as harmful substances, in turn making them (safe) easy to access, leads to many people misusing them without regard for their potential negative effects, to the point where they have become the most common cause of poisoning in Latin countries [9]. The influence of religious factors and cultural customs in society reduces or eliminates intended drug poisoning to the point where negligent consumption of substances such as painkillers (Panadol) may now be the most important cause of drug poisoning [10,11]. However, apart from drug toxicity, toxicity associated with the daily use of chemicals has increased dramatically because of a lack of understanding of how to handle chemicals used daily. The use of chemicals in many industrial, medical, and agricultural fields has increased substantially, owing to their growth and development. Every year, more than two million people die as a result of acute poisoning caused by chemicals. Chemical poisoning frequently occurs by accident, particularly in children [4]. Chemical poisoning is dangerous because it is acute poisoning, meaning that it manifests with a rapid, severe effect in less than 24 h [12]. Chemical poisoning has been reported in many Saudi cities, including Jeddah, Riyadh, Makkah, Najran, and Al-Qassim, in various age groups due to a variety of factors [13,14].

Overall, the effect of toxicity cases, whether from food contamination, prolonged drug administration, or chemical mishandling, varies with various factors such as age, nature of immunity, poison reception, toxicity estimation, and administration. In the past couple of decades, Saudi Arabia has achieved major milestones in agriculture, industrial technology, and medical pharmacology. These advancements have resulted in the development and widespread availability of a wide range of toxic agents, including pesticides, therapeutic drugs, and other chemicals, as well as significant changes in the trends of acute poisoning [10]. Toxic agents linked to morbidity and mortality, as well as the pattern of acute poisoning, which varies from place to place and over time, are expected to change. Therefore, it is necessary to procure up-to-date information on acute poisoning to plan rational resource allocation and evaluate public health interventions. The objectives of this retrospective and descriptive study were to characterize the clinical and sociodemographic patterns of acute poisoning in the emergency department (ED) of King Fahad Hospital and poison centers in Saudi Arabia’s Al-Baha province. Subgroups of toxic agents were classified based on their intended use. The ingestion of two or more drugs was defined as a mixed drug. If no suspected toxic agent was reported in the patient’s history, the toxic agent was classified as unknown. In the case of food poisoning, the type of exposure is classified as food contamination by various microorganisms at home, restaurants, or weddings. Suicidal, accidental, and unknown drug and chemical toxicity were classified as follows: (1) Suicidal exposure resulting from the inappropriate use of drugs for self-destruction. (2) Abuse was defined as exposure due to intentional improper or incorrect use of a drug in which the victim was likely attempting to achieve a euphoric or psychotropic effect. (3) Unintentional poisoning was defined as environmental poisoning, misuse, food poisoning, or adverse reactions. Gastric lavage, activated charcoal, and other antidotes and antibiotics are recommended for patients with food, chemical and drug toxicity.

## 2. Materials and Methods

### 2.1. Methodology and Data Collection Analysis

This descriptive cross-sectional study and retrospective case-series study were performed to assess the various poisoning cases, particularly food, drug, and chemical poisonings. Data were collected from tertiary hospitals and poison centers located in Al-Baha Province in the Kingdom of Saudi Arabia. From 2019 to 2022, 622 poisoning cases were collected during the study period. We identified 159 patients with food poisoning, 377 patients with drug toxicities, and the remaining 86 patients with chemical poisoning out of the total number of poisoning cases. This study enrolled records of patients who complained of mild, average and serious symptoms of drug, chemical, and food poisoning. According to Persson’s poison severity score, individuals were classified as having mild, moderate and severe/serious cases of poisoning (Table 1).

Patient histories, physical examinations, and routine toxicological laboratory tests were used to make the diagnoses. An electronic hospital information system was used to obtain the relevant medical records. In a clinic, an electronic system is used to record patient information. Searching for some of the following keywords yielded various food, chemical, and drug poisoning cases: food poisoning, chemical and home-based chemical poisoning, and drugs/medication adverse effects. A structured form was used to collect and document demographic data such as age, sex, place of residence, diagnosis, type of exposure (suicidal, abusive, accidental, and unknown), type of toxic agent (the common name was indicated, where available), and treatment outcome (whether the patient survived or died). Therapeutic drugs, pesticides, alcohol, poisonous fumes/gases/vapors, chemicals, food, other substances, and unknown compounds were classified as toxic agents. Symptomatic treatment was administered in the absence of specific antidotes. As a result, the toxic agents linked to fatalities and morbidity, as well as the pattern of acute poisoning, which varies from place to place and over time, are expected to change. There is a constant need to obtain updated information on acute poisoning to plan rational resource allocation and evaluate public health interventions. Epidemiological studies are urgently needed in each country and region to clarify the profile of acute poisoning and assist authorities in developing a plan for its prevention and control. Numerous studies have been conducted on acute poisoning in various Saudi populations. To our knowledge, no study on poisoning cases, whether food, chemical, or drug, has been conducted in Al-Baha City from 2019 to 2022.

### 2.2. Statistical Analysis

The data were analyzed using the SPSS/PC statistical program. Simple frequency tables, as well as means and standard deviations, were used to generate descriptive statistics. Chi-square tests were used to compare categorical variables in inferential statistics. The statistical significance level was set at *p* < 0.05.

### 2.3. Patient and Public Participation

There was no involvement of patients or the general public.

## 3. Results

### 3.1. Aspects of Food Toxicity Instances in the Al-Baha Region

#### 3.1.1. Demographic Variables

A total of 159 cases of food poisoning were reported to the toxicology center of King Fahad Hospital and the poison center regions of Al-Baha from 2019 to 2022. Based on the report of food poisoning, the highest percentage of poisoning was observed in males (85 cases) whereas 74 cases were female patients. The lowest age of the patient was found to be a one-year-old kid and highest age of the patient that was suffering from food poisoning was 89 years old. The patients most vulnerable to poisoning were those aged 1–10 years (105 cases) followed by patients aged 11–20 (17 cases) and above 40 years (16 cases) (Table 2).

Furthermore, data analysis revealed that *Salmonella* was the most common causative agent of infection (149 cases), followed by six cases that contained the *Enterococcus* species and four cases that contained *E. coli*. *Salmonella* infection can sometimes be caused by a variety of meat products, such as chicken, beef and eggs, fruits, sprouts, various vegetables, and even manufactured foods such as nut butters, chicken nuggets, and stuffed chicken meals [10]. Because *Enterococcus* poisoning is less common among patients, hygienic conditions in the Kingdom are not jeopardized. Furthermore, *E. coli* infections are less prevalent in Al-Baha, implying that people do not consume more contaminated raw food, particularly raw meat and vegetables. Based on the geographic distribution of the cities in the Baha province, the city of Al-Baha comes in with the highest rates of food poisoning, with 109 subjects (68.5%) followed by 24 (15.1%) in Al-Agig. The incidence rate of food poisoning in Almandag, Algara, Bhaljureshi and Qilwa are ten (6.3%), six (3.8%) and four cases (2.5%), respectively. However, 136 (85.5%) poisonings associated with foodborne diseases were reported at home, whereas 23 (14.4%) cases were reported from the restaurants and weddings. Depending on the severity of the infection as determined by signs and symptoms and grading score, 67 patients were diagnosed with mild infection (42.1%) followed by 63 patients (39.6%) with an average infection. Serious infection was observed in 29 (18.3%) patients. The treatment regimens used in our study included antibiotics and supportive care.

Innumerable antibiotics were administered according to the age groups. We observed that 78 patients were administered with Augmentin antibiotics to treat foodborne infections, with a percentage of 49.15%. Quinolones were then administered to 35 patients, with a success rate of 22.0%. Cephalosporins were administered as a therapeutic regimen in 29 patients (18.2%). Penicillin antibiotics were recommended to 17 (10.7%) patients. 

Depending on the signs and symptoms patients encountered during food poisoning, health care practitioners have discovered various diagnostic tools. Approximately 150 patients (94.3%) complained of common signs and symptoms such as diarrhea, fever, and stomach cramps. Six patients (3.8%) had severe signs and symptoms such as fever with chills, severe headache, severe abdominal pain, micturition, and nausea. Three patients (1.9%) reported diarrhea, stomach cramps, and occasional fever. According to a report on food poisoning from 2019 to 2022, the highest poisoning levels were documented in 2021, 68 (42.8%), followed by between 2019 and 2022, 34 (21.4%), and 2020, 23 (14.5%) (Table 2).

#### 3.1.2. Correlation between Infection Severity, Type of Organism Involved, Signs and Symptoms, and Antibiotics Prescribed by Age Group

Table 3 lists the types of organisms with the severity of infection, along with the treatment regimen. A significant difference (*χ*^2^ = 44.345 and *p* < 0.05) was observed between the various age groups of patients with respect to the type of organism causing food poisoning. Various bacterial infections were reported among patients of different ages starting from one year to 89 years old. It was noticed that around 104 cases of *Salmonella* were found in patients of an age group of between 1 and 10 years. Between patient ages of 11- and 20-years age it was noticed that only 17 patients had food poisoning because of *salmonella* contamination. Twenty-two Patients aged 31–89 years were infected with *Salmonella*, and six patients aged 21–30 years were infected with *Salmonella*. *Enterococcus* infection was rarely seen among patients suffering from food poisoning, as only two patients within the age of 21 to 40 were infected, and four patients aged ≥40 y had *Enterococcus*. This means that the infection was observed to be highest in children aged between 1 and 10 years. However, they may not be resistant to infection. Necessary precautionary measures should be taken to ensure the safety of children in this age group. A statistically significant difference (*χ*^2^ = 21.123, *p* < 0.001) was observed in the severity of infections at different stages with respect to patient age. Patients aged between 1 and 10 years exhibited all stages of infection compared with other age groups (11–20 y, 21–30 y, 31–40 y and ≥40 y). Forty-three patients aged 1–10 years, nine patients aged 11–20 years, four patients aged 21–30 years, two patients aged 31–40 years and five patients aged > 40 years had an average infection rate. Similarly, 41 patients aged 1–10 years, 8 patients aged 11–20 years, 5 patients aged 21–30 years, 7 patients aged 31–40 years and 6 patients aged > 40 years had mild infections. A total of 20 patients aged 1–10 years, 3 patients aged 11–20 years, 0 patients aged 21–30 years, 3 patients aged 31–40 years and 3 patients aged > 40 years had serious infections (Figure 1). People at high risk of food poisoning included children between the ages of 1 and 10, compared to other age groups. Due to the fact that their immune systems are still growing, and they are unable to fight infections as well as older children and adults, children under the age of 15 years have a significant risk of foodborne illnesses and other related health issues. A statistically significant difference (*χ*^2^ = 103.678, *p* < 0.001) was observed in the administration of antibiotics to various patients in different age groups. Augmentin is a mild antibiotic that was prescribed to 73 patients aged till 1 –10 years. Only five patients within the age group of 11–20 years were recommended Augmentin antibiotics. For older age groups (21–89 years), Augmentin was not recommended to the patients. A total of 73 patients were prescribed Augmentin, 21 patients were prescribed cephalosporins, 4 patients were prescribed quinolones and 7 patients were prescribed Penicillin within the age group 1 to 11 years of age. Similarly, five patients received Augmentin, five received cephalosporins, four received quinolones, and three received Penicillin from 11 to 20 years of age. Similarly, six patients received quinolones, and three received Penicillin from 21 to 30 years of age. Additionally, eleven patients received quinolones and one received Penicillin in the 31 to 40 years age group; three received cephalosporins, ten received quinolones, and three received Penicillin in the >40 years age group (Figure 2). A statistically significant difference (*χ*^2^ = 103.678 and *p* < 0.001) was observed. Most cases had symptoms of diarrhea, fever, and stomach cramps in children aged one to eleven years, and severe signs and symptoms, such as fever with chills, severe headache, severe abdominal pain, micturition, and nausea, was observed only in a few patients (Table 3).

### 3.2. Aspects of Drug Toxicity Instances in the Al-Baha Region

In accordance with the report, drug poisoning accounted for the majority of poisoning-related admissions with a total case of 377.

In the present study, the incidence rate of drug poisoning in males was 204 (54.1%), while in females it was 173 (45.9%), and the male to female ratio was 1.2:1. The patients most vulnerable to poisoning were those aged 1–10 years (*n* = 120, 31.8% of cases), followed by patients aged 11–20 years (*n* = 101, 26.8% of cases). In the age group of 21–30 years, 74 patients (19.6%) were vulnerable to various kinds of drug toxicities. A total of 47 cases of drug toxicity were found in the age group of 31–40 years old with a 12.5% vulnerability rate, and 35 patients (9.3%) were above 40 years of age. Table 4 shows the frequency of, and percentage of the patients with, adverse drug effects. It was noticed that out of 377 total drug poisoning cases, 363 cases with a percentage of 96.3% were accidents. This indicates that these were unintentional cases. Patients were unaware of their adverse effects. The remaining 14 patients (3.7%) had suicidal ideations. These cases were intentional in nature. Patients knew the adverse effects of the drugs and they intentionally consumed the drugs for suicidal purposes.

Among the various routes of exposure, ingestion (oral) was the most common (92.83%), followed by inhalation (nasal) (5.30%) and other unknown routes (1.87%). Various drugs caused different adverse effects when administered depending on the severity of the infection. Furthermore, we found that out of 377 patients with drug toxicity, 376 recovered and one died. Of these, 360 were Saudi and 17 were non-Saudi. In 2019, 146 patients (38.7%) suffered from drug poisoning symptoms; in 2020, 76 (20.2%) had drug poisoning; in 2021, 122 patients (32.4%) had drug poisoning; and in 2022, 33 patients 33 (8.8%) had adverse drug effects

The most prevalent drugs and other therapeutic drugs that cause severe adverse events are shown in Table 5 and are listed by frequency of exposure in various age groups of study patients. The most common toxic agent groups were various therapeutic drugs and other drug abuse substances consumed by patients. A total of 377 cases involved therapeutic drug poisoning. Of these, 66 (17%) involved analgesic poisoning, which was the most commonly used drug, followed by anticonvulsants (5.6%) (including 20 cases of antipsychotics). Other categories of unknown drugs constituted approximately 30.3% of the drugs, but their origin is unknown. Antitussive, antispasmodic, antimicrobial, traditional medicine and other drugs showed a relatively stable trend. Table 5 shows the frequency of various therapeutic drugs and other drugs consumed by various patients.

Drug poisoning was common in both men and women aged 10–30 years. However, drug poisoning was more common in men, with a total number of 204 cases, compared to women, which were 173 total cases in number. There was a statistically significant difference noticed between gender and the type of the drug they consumed, with a chi square value *χ*^2^ 77.301 and *p* value < 0.001. Patients of different age groups were investigated to determine impact of various drugs. A statistically significant difference was observed between different age groups and drug toxicity. Table 6 shows the impact of therapeutic and other drugs in with different age groups.

It was noticed that various therapeutic drugs used to treat various disorders had predominantly caused more adverse effects in those aged between 1 and 20 years of age. However, some recreational drugs (drugs of abuse) were recently reported to cause adverse effects in patients aged 21–30 years old. Precautionary measures should be taken to stop the trend of severe adverse effects when using recreational drugs. A statistically significant difference was found, with patients showing notable variations in their age with respect to the drugs administered. A chi value *χ*^2^ of 215.236 was observed, and a significant difference (*p* < 0.05) was found between the different age groups. It was noted that the administration of N-acetyl cysteine and activated charcoal were administered in 39.33% (*n* = 83) and 30.06% (*n* = 65) of cases, respectively, to prevent the absorption of toxic agents. In addition, 6.6% of the patients received naloxone as an antidote, 5.76% received omeprazole, 6.16% received sodium bicarbonate, 4.735 received glutamate (Figure 3).

Most patients complained of severe signs and symptoms such as abdominal movement, dystonia, sleeping disorders and epigastric pain. These symptoms were followed by other serious symptoms, such as headache, nausea, vomiting, dizziness, confusion spasm and disorientation. Other symptoms that were observed at lower frequencies in patients were watery diarrhea and seizures.

### 3.3. Aspects of Chemical Toxicity Instances in the Al-Baha Region

Despite a significant number of cases associated with food and drug poisoning, many cases of chemical poisoning were recorded. The current study revealed that 64 (74.4%) were in males and 22 (25.6%) were in females, and the male to female ratio was 3:1. Almost 48% (*n* = 41) of all chemical poisoning cases involved children in the group of 11–20 years old. This group was followed by 1–10-year-old children, whose contribution was 42% (*n* = 36) (Table 7). In the age group of 21–30 years old, around eight patients (9.3%) had chemical toxicity; very few exposed patients were in the age group above 40 years (*n* = 1). Methanol toxicity along with unknown drug administration was the most common chemical poisoning agent observed, with a frequency of 21 patients constituting 24.4% of the total 100%. Among the various cleaning agents, Clorox was the second most common cause of chemical poisoning in 16 patients (18.7%). Spraying of insecticides and pesticides also caused severe chemical poisoning, which was observed in 15 patients constituting 17.2% of the aggregate. In addition, alcohol poisoning (10.5%), hair dye poisoning (3.5%), and other chemicals were also observed (Table 7). In the group of construction or building materials, thinners contributed the majority of poisoning cases. Figure 4 shows the percentage of various chemical toxicants causing chemical toxicity in patients.

Among the various routes of exposure, swallowing was the most common (68.6%) route followed by inhalation (nasal) (24.4%). The dermal and injection routes constituted approximately 4.75 and 2.3%, respectively. Various drugs caused different adverse effects when administered depending on the severity of the infection. In our study, we noticed that among 76 patients, 42 (48.8%) had average infection, 28 (32.5%) had serious infection, and 16 (18.65%) had mild infection. Of the 86 patients with chemical toxicity, 85 recovered and one died. Of these, 75 were Saudi and 11 were non-Saudi. In 2020, 37 (43%) patients had chemical poisoning; in 2022, 19 patients (22.1%) had chemical poisoning; in 2022, 15 patients (17.4%) had chemical adverse effects; and in 2021, 15 patients (17.4%) suffered from chemical poisoning symptoms.

#### Distribution of Toxic Agents with Respect to Age

A statistically significant difference was observed between sex and chemical toxicant exposure, meaning that males were more exposed to toxic chemicals than females. In addition, we also correlated the ROA with males and females and found that males were exposed to chemical contaminants more through swallowing than females at ratio of 5:1, followed by inhalation (1:1), dermal (2:1) and injection (1:1). The chi- square value was *χ*^2^ = 8.294 and *p* value < 0.05. A significant value was observed in the outcome to exposure with respect to gender (*χ*^2^ 8.25, *p* value < 0.01). We found that 63 male cases out of 86 were accidental and only one was suicidal, compared to 18 female cases of accidental chemical exposure and four suicidal. Various therapeutic regimens were administered in the form of antidotes, as shown in Table 8, with *χ*^2^ = 45.94 and a *p* value < 0.001

A statistical difference of 0.001 and chi square value of 28.038 was noted between the outcome of chemical poisoning and the age groups of various patients, and it was noted that on average infection was most commonly seen in patients with mild and serious chemical toxicity. Furthermore, a statistical significance of *p* < 0.01 with a *χ*^2^ value of 82.171 was found in patients when the type of toxicant was corelated with various age groups. Moreover, only five suicidal cases were found between 1 and 20 years of age. Almost all cases of chemical poisoning were accidental. A statistically significant difference was also observed in the route of administration and treatment regimen when their effect was observed in various age groups (Table 9).

## 4. Discussion

One of the main issues regarding public health in this nation is poisoning. Information on the type and severity of poisoning is required to initiate preventive actions [16]. Poisoning continues to pose a serious and expanding hazard to public health worldwide, significantly contributing to hospital admissions, morbidity, and mortality. The prevalence of food poisoning varies from city to city and from area to area. Food poisoning is a substantial contributor to both short- and long-term morbidity; it also contributes to both overt and covert morbidity [17]. Only a few epidemiological studies on poisoning have been published in various regions of Saudi Arabia, including Qassim, Makkah, and Jeddah [4]. This study was conducted to evaluate the pattern of poisoning in Al-Baha, Saudi Arabia for the first time. To the best of our knowledge, there have been no epidemiological studies on poisoning in Al-Baha City that might be utilized to teach the general public about the substances that cause poisoning most commonly in this area and about the precautions that should be taken to avoid them. These studies may also inform doctors about the risk factors and epidemiology of poisoning, and they may offer guidance on how to treat poisoning as well as guidelines for avoiding it. 

Moreover, results from various countries, especially Saudi Arabia, show that children under the age of five were the most frequently exposed to drug, food and chemical toxicity. According to these results, Saudis account for the majority of the drug poisoning cases, which is consistent with earlier Saudi research [11]. Additionally, accidental poison intake was the reason for the majority of instances in the research. The results of the current study showed considerable variation in poisoning in the Kingdom of Saudi Arabia. There were no discernible causes other than the disparity between the population and the number of eateries. Surprisingly, the number of poisoning cases reported at home appeared to be larger than the number of cases recorded in restaurants in the current study. Restaurants are a significant global source of tainted food and foodborne illnesses. Food, pharmaceutical home goods, and plant product derivatives with varying degrees of toxicity can be characterized as ingested toxins (low, intermediate or highly toxic) [18]. Acute poisoning is one of the most prevalent causes of hospital admissions and death. Food poisoning, also known as microbial foodborne illness, is a prevalent occurrence worldwide including Saudi Arabia [19]. Fortunately, the majority of people recover from foodborne disease episodes without the need for medical attention, or long-term consequences. Foodborne sickness can be caused by a wide range of bacteria. The most common bacterial cause of foodborne diseases worldwide is Salmonella, which is also increasing in prevalence each year in Saudi Arabia [20]. The results of the current study showed that *Salmonella* is responsible for various food-related complications in various age groups, especially in children [21]. This study also found that, especially among the age range from one year to ten years old and with a female to male ratio of 1.2:1, men were somewhat more likely to experience poisoning than women. This is supported by a previous study [22] showing the prevalence of male foodborne infections to be comparatively higher than female patients. Food poisoning is caused by various bacterial contaminations; in our study, we found that patients infected with *Salmonella* outnumbered *Enterococcus* and *E. coli* infections among patients [23]. This study was supported by a recent study conducted by Qi et al., in 2019, which showed the occurrence of *Salmonella* food poisoning [24]. Furthermore, our results showed that food poisoning caused by bacterial contamination is mild and not serious. Only a few cases were found to be serious, for which necessary intervention and therapeutic regimen was provided, such as various antibiotics, depending on the age of the patients. In line with our findings, a recent study showed the prevalence of foodborne infections in the country among patients, and how necessary interventions and pharmacotherapy helped patients to recover [25]. Similarly, the current increase in food poisoning is attributable to a variety of food sources, including restaurants and food trucks, as well as poor cleaning practices [26]. Likewise, drugs and chemical toxicity are prevalent in various regions of Saudi Arabia. As we reported for the first time in Al-Baha, we achieved a substantial outcome by showing poisoning prevalence in diverse populations. Furthermore, the results of the current study indicate that incidences of food, chemical, and drug poisoning show many variations among all age groups. We found consistency in food poisoning cases among children, particularly in the 1–20-year patient age group. Our study is consistent with previous studies of food, drug, and chemical poisoning cases [26]. It has been claimed that male poisoning cases outnumber female poisoning cases in other nations, which goes against the study’s findings. According to findings from earlier research conducted in Saudi Arabia, women are more prone than men to attempt suicide and poison themselves with medicines. According to the results, Saudis accounted for the majority of drug poisoning cases, which is consistent with earlier Saudi research. Additionally, in the majority of instances in this research, accidental poison intake was the reason. A similar pattern was also observed in other studies [27]. Contrary to the findings of this study, the most common drug poisoning reason was suicide [28]. Additionally, according to the results of the current study, analgesics were the most frequently used medications involved in pharmaceutical drug poisoning, which is consistent with a study from Saudi Arabia. Adults and children most often use analgesics and NSAIDs, which is consistent with the majority of worldwide and national research [29]. However, according to the results of the current study, men were more likely than women to experience chemical poisoning, which is consistent with the study published by. Contrary to our findings, several studies have claimed that male preponderance in poisoning was found in the province of Al-Baha. In addition, children between the ages of 1 and 20 years were the age group most frequently exposed to chemical poisoning, which is consistent with data from prior Saudi research in Riyadh and Qassim. In contrast to our findings, a recent study revealed that individuals between the ages of 21 and 30 comprised the majority of chemical poisoning cases [30]. The results also revealed that Saudis made up the bulk of the instances of chemical poisoning, which is consistent with earlier Saudi investigations [4]

Additionally, unintentional chemical poisoning has been reported in several studies [31]. It is interesting to note that the results of this investigation showed that the administration of methanol and unidentified medicines was linked to chemical poisoning. A study conducted by Brent in 2001 highlighted the high frequency of chemical poisoning among children [32]. Our study also revealed that the most potent bleach (Clorox^®^, Oakland, CA, USA) was the most prevalent home product implicated in chemical poisoning. Among the recorded instances, activated charcoal was the most effective treatment method. It is well recognized that administering activated charcoal to patients who have recently overdosed on organochlorine pesticides and herbicides considerably limits their total medication absorption. Additionally, it can be used to reduce the absorption of toxic and narcotic overdoses through gastrointestinal decontamination. The oral route is the most prevalent type of exposure reported in the present study, and it is conceivable that this was because patients between the ages of 11 and 20 years old were those who reported the most occurrences of unintentional medication ingestion. The results of this study also showed a strong correlation between the ages of those in the study and the types of agents used in poisoning cases. However, despite some of the prior literature supporting this conclusion, several other studies contradict it. Additionally, because no consistent pattern can be observed, it is challenging to pinpoint the cause of these changes using the number of instances for each type. Furthermore, our investigation showed that the most common toxicants used were prescription medications. In both industrialized and developing nations, medicinal medications have been shown to follow a similar trend as the most frequent cause of acute poisoning. Sedatives/hypnotics are often consumed in Taiwan, Hong Kong, Iran, and Finland with regards to the subgroup of therapeutic medications in poisoning cases, whereas analgesics are most frequently consumed in Turkey and the USA [33]. The majority of drug poisonings observed in the current study were caused by pharmaceuticals, of which analgesic medications (17%), unknown drugs (30.5%), anticonvulsants, and antipsychotics were the most frequently consumed substances. However, a change from this previously recognized trend was noted in the use of analgesics, cold and cough medications, combination pharmaceuticals, and hypertensive drugs. Additionally, we discovered an increase in the incidences of psychotropic medication poisoning, which suggests that mental illness is a factor in poisoning incidents and needs to be addressed [34]. However, an increase in the use of analgesics, cold and cough remedies, combination pharmaceuticals, and hypertensive medications was observed, resulting in a pattern different from that previously reported [35]. Additionally, we discovered that the frequency of psychotropic medication poisoning has grown, indicating that mental illness may contribute to poisoning occurrences and that it is important to pay attention to this. Surprisingly, there were few opioid poisonings in our study, which is related to the strict regulation of opioids and physicians’ and patients’ lack of knowledge about opioids (such as excessive worry about addiction and negative effects), making the use of opioids uncommon. The main limitation of our study stems from the fact that it was retrospective and hence patient data was missing. Another drawback is that, despite having three branches, the data from a single teaching hospital may not accurately represent the circumstances in this area. Therefore, thorough data gathering and analysis from other general hospitals in the area may more precisely portray the patterns of regional poisoning. Additionally, the lack of data standardization made comparison between the two studies shaky.

## 5. Conclusions

These findings add to our understanding of the prevalence and incidence of acute poisoning in Al-Baha. The results revealed that more men than women experienced acute poisoning, with young adults aged 1–20 years being the most vulnerable. We also discovered that the vast majority of poisonings were unintentional, with very few intentional poisonings. The results of the current study showed that pharmaceutical and chemical items were the primary causes of poisoning in Al-Baha. According to our findings, therapeutic drugs were the most commonly implicated group of toxic agents, followed by pesticides and alcohols, particularly methanol. Poisoning from analgesics, anticonvulsants, and antipsychotics was more common than poisoning from other therapeutic drugs. Furthermore, the incidences of disinfectant poisoning have significantly increased. Based on the findings of the current study, the following recommendations are made. Firstly, vulnerable groups, such as young adults, require special attention due to their limited coping abilities. To reduce acute poisoning, more investment is needed to promote public health education on the rational use and safe storage of toxic agents, as well as self-protection. Patients who attempt suicide should seek psychiatric help as soon as possible. In suicide patients, early psychiatric consultation and identification may reduce the risk of further self-harm. In general, relevant policies and regulations should be developed and implemented as soon as possible to limit access to toxic agents, particularly pesticides, owing to their high toxicity. Furthermore, to prevent the high incidences of poisoning among children and adolescents, parental awareness should be increased, and educational programs should be made available to the general public.

## Figures and Tables

**Figure 1 healthcare-11-01398-f001:**
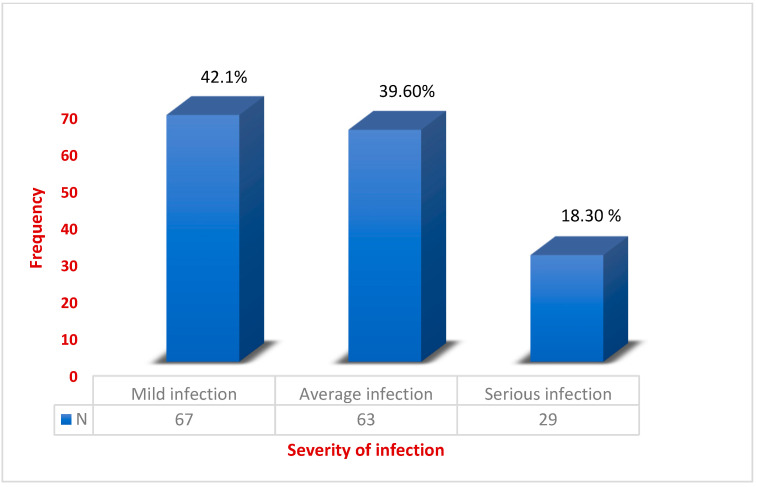
The severity of infections among patients based on the Persson severity score scale (*n* = number of patients and % signifies percentage).

**Figure 2 healthcare-11-01398-f002:**
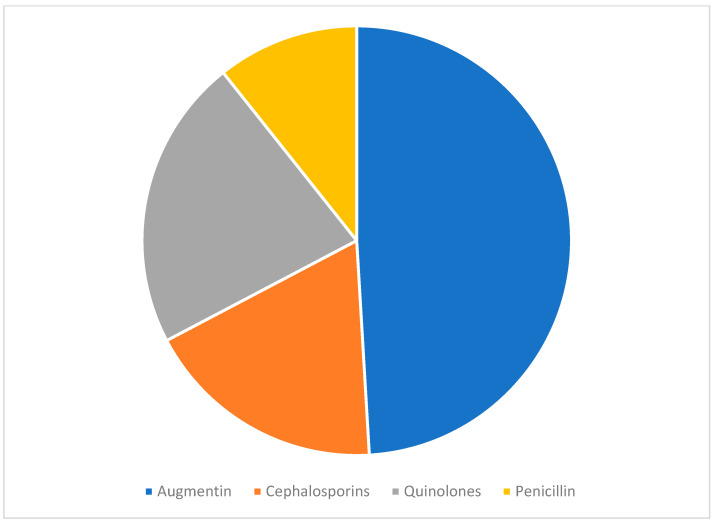
Showing the range of antibiotics prescribed depending on the age group of patients (1 (Augmentin); 2 (Cephalosporins); 3 (Quinolones) and 4 (Penicillin).

**Figure 3 healthcare-11-01398-f003:**
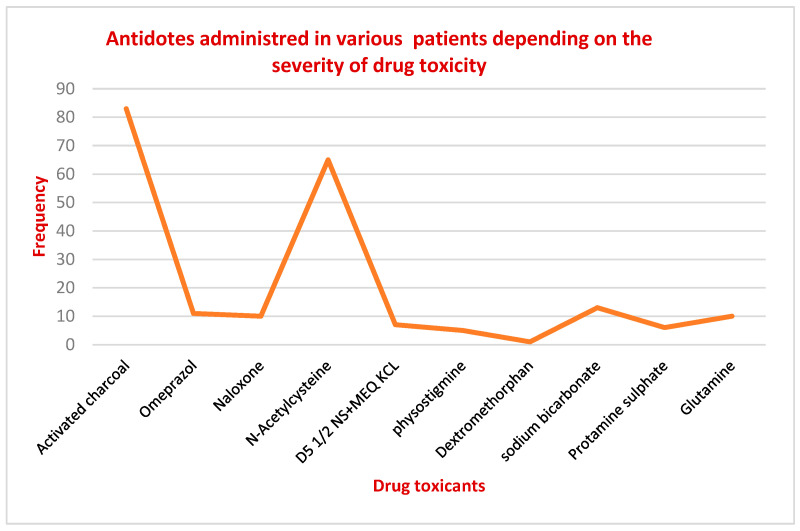
Various antidotes administered to patients.

**Figure 4 healthcare-11-01398-f004:**
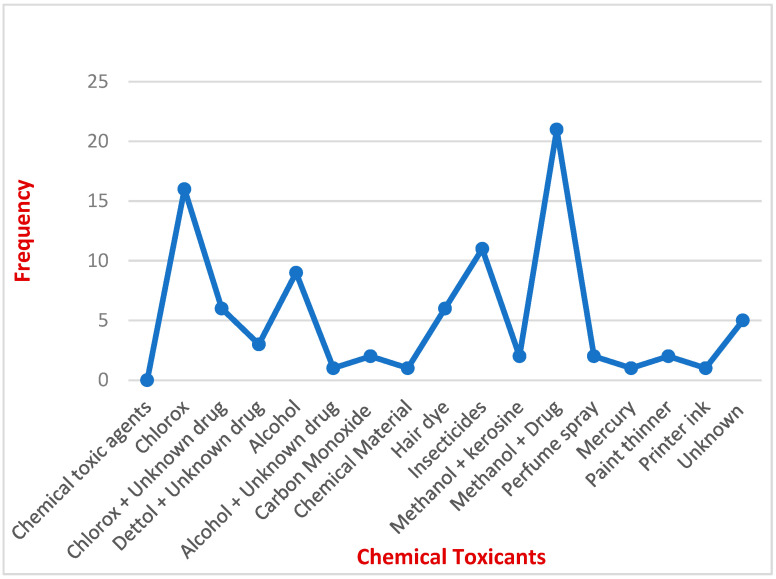
Various chemical toxicants associated with various toxicities.

**Table 1 healthcare-11-01398-t001:** Persson et al. poisoning severity scores [15].

Severity Grade Score	Symptoms
None 0	No symptoms or signs related to poisoning
Minor 1	Diarrhea, fever, stomach cramps
Moderate 2	Diarrhea, stomach cramps, occasional fever
Severe 3	Fever with chills, severe headache, severe abdominal pain, micturition, nausea
Fatal 4	Death

**Table 2 healthcare-11-01398-t002:** Demographic characteristics of various patients suffering from food poisoning.

Age Group	*n*	%
1–10	105	66%
11–20	17	10.7%
21–30	9	5.7%
31–40	12	7.5%
≥40	16	10.1%
Gender		
Male	85	53.5%
Female	74	46.5%
Organism Type		
*Salmonella*	149	93.7%
*Enterococcus* species	6	3.8%
*E. coli*	4	2.5%
Region		
Albaha	109	68.5%
Qilwa	4	2.5%
Alaqiq	24	15.1%
Algara	6	3.8%
Almandag	10	6.3%
Bhaljureshi	6	3.8%
Place of Poisoning		
Home	136	85.5%
Restaurants and weddings	23	14.5%
Severity of Infection		
Average infection	63	39.6%
Mild infection	67	42.1%
Serious infection	29	18.3%
Type of Antibiotics		
Augmentin	78	49.1%
Cephalosporins	29	18.2%
Quinolones	35	22.0%
Penicillin	17	10.7%
Year		
2019	34	21.4%
2020	23	14.1%
2021	68	42.8%
2022	34	21.4%
Signs and Symptoms		
Diarrhea, fever, stomach cramps	150	94.3%
Fever with chills, severe headache, severe abdominal pain, micturition, nausea	6	3.8%
Diarrhea, stomach cramps, occasional fever	3	1.9%

**Table 3 healthcare-11-01398-t003:** Type of exposure of all the cases with therapeutic regimen by age.

Severity of Infection	1–10 y	11–20 y	21–30 y	31–40 y	≥40 y	*χ* ^2^	*p* Value
Average infection	43	9	4	2	5	21.123	**0.001 ^s^**
Mild infection	41	8	5	7	6		
Serious infection	20	3	0	3	3		
Organism							
*Salmonella*	104	17	6	11	11	44.345	**0.05 ^s^**
*Enterococcus* species	0	0	1	1	4		
*E. coli*	1	0	2	0	1		
Signs and Symptoms							
Diarrhea, fever, stomach cramps	104	17	6	11	12	48.392	**0.001 ^s^**
Fever with chills, severe headache, severe abdominal pain, micturition, nausea	0	0	1	1	4		
Diarrhea, stomach cramps, occasional fever	1	0	2	0	0		
Antibiotics Prescribed							
Augmentin	73	5	0	0	0	103.678	**0.001 ^s^**
cephalosporins	21	5	0	0	3		
quinolones	4	4	6	11	10		
Penicillin	7	3	3	1	3		

*χ*^2^ = chi-square analysis; s = significant difference; ns = non-significant difference; significance of *p* values is in bold.

**Table 4 healthcare-11-01398-t004:** Demographic characteristics of drug poisoning among patients.

Variables Categories	Frequency	%
Age Group		
1–10	120	31.8%
11–20	101	26.8%
21–30	74	19.6%
31–40	47	12.5%
≥40	35	9.3%
Gender		
Male	204	54.1%
Female	173	45.9%
Type of Exposure		
Accidental	363	96.3%
Suicidal	14	3.7%
Route of administration		
Ingestion (Oral)	350	92.83%
Inhalation (Nasal)	20	5.30%
Other	7	1.87%
Place of Residence		
Al-Aqiq	4	1.1%
Al-Baha	216	57.3%
Al-Gara	17	4.5%
Al-Hajra	2	0.5%
Al-Mandaq	2	0.5%
Al-Miha	1	0.3%
Al-Mikhwah	12	3.2%
Biljurashi	103	27.3%
Namerah	1	0.3%
Qilwa	19	5.0%
Severity of infection		
Mild infection	97	25.7%
Average infection	217	57.6%
Serious infection	63	16.7%
Outcome		
Recovered	376	99.7%
Death	1	0.3%
Nationality		
Saudi	360	95.5%
Non-Saudi	17	4.5%
Year of Drug Poisoning		
2019	146	38.7%
2020	76	20.2%
2021	122	32.4%
2022	33	8.8%

**Table 5 healthcare-11-01398-t005:** Distribution of drug toxicants consumed by male and female patients.

Type of Drugs	*n*	%	Females	Males
Analgesic	66	17	39	27
Antiarrhythmics	1	0.3	1	0
Antibiotics	9	2.4	1	8
Anticoagulants	8	1.1	5	3
Anticonvulsants	20	5.6	6	14
Antidepressants	25	5.9	13	12
Antidiabetic	2	0.6	1	1
Antiemetic	2	0.5	2	0
Antihistamine	5	1.5	3	2
Antilipemic	1	0.3	1	0
Antipsychotics	8	1.1	2	6
Antispasmodic drug	1	0.3	1	0
Antitussives	5	4.1	0	5
Aspirin	10	2.7	2	1
Atypical Antipsychotic	15	2.8	8	7
Beta agonists	1	0.3	1	0
Biguanides	4	1	4	0
Caffeine	3	0.9	3	0
Beta agonists	1	0.3	1	0
DPP-4 inhibitors	1	0.3	1	0
Drug of abuse	3	0.9	0	3
Fenethylline	1	0.3	0	1
Hormones	3	0.9	1	2
Hypertensive drugs	4	1	1	3
Laxatives	4	1	3	1
Leukotriene receptor antagonists	1	0.3	1	0
Multiple drug ingestion	17	4.5	13	4
Mycophenolic acid	2	0.3	2	0
NSAIDs	12	3.1	7	5
Opioid analgesics	6	0.7	2	4
Contraceptives	2	1	0	1
PDE5 inhibitors	1	0.3	0	1
Pentagon	1	0.3	1	0
Phenethylamine	13	3.4	12	1
Poisonous herbs	1	0.3	0	1
Proton pump inhibitors	1	0.3	1	0
Rythmodan	1	0.3	0	1
Skeletal muscle relaxants	4	2	2	2
Sulfonylurea	2	0.6	2	0
Teriflunomide	1	0.6	1	0
Tetrahydrocannabinol	1	0.3	0	1
Unknown	113	30.5	34	79
Vitamins	5	0.8	3	2
Xanthines	1	0.3	1	0

*χ*^2^ = chi-square analysis (77.301); *p* < 0.001 indicates that the differences found were statistically significant.

**Table 6 healthcare-11-01398-t006:** Type of drug toxicities reported according to the patients age groups.

Type of Drug Toxicity	1–10	11–20	21–30	31–40	≥40
Analgesic Antiarrhythmics	18 1	25 0	16 0	5 0	2 0
Antibiotics	2	3	1	1	2
Anticoagulants	5	1	0	2	0
Anticonvulsants	5	4	5	3	3
Antidepressants	0	6	2	14	3
Antidiabetic	0	1	0	1	0
Antiemetic	1	0	0	1	0
Antihistamine	5	0	0	0	0
Antilipemic	1	0	0	0	0
Antipsychotics	3	1	2	1	1
Antispasmodic drug	0	0	1	0	0
Antitussives	3	1	1	0	0
Atypical Antipsychotic	6	5	3	1	0
Beta agonists	0	0	1	0	0
Biguanides	1	2	1	0	0
Caffeine	1	2	0	0	0
Contraceptives	0	0	2	0	0
DPP-4 inhibitors	0	1	0	0	0
Drug of abuse	0	0	2	1	0
fenethylline	0	0	1	0	0
Hormones	2	0	0	1	0
Hypertensive drugs	0	1	3	0	0
Laxatives	3	1	0	0	0
Leukotriene receptor antagonists	1	0	0	0	0
Multiple drug ingestion	4	8	4	0	1
Mycophenolic acid	1	0	0	0	1
NSAIDs	3	4	3	2	0
Opioid analgesics	0	0	2	3	1
PDE5 inhibitors	0	0	0	1	0
Pentagon	0	0	1	0	0
Phenethylamine	0	3	4	5	1
Poisonous herbs	1	0	0	0	0
Proton-pump inhibitors	0	0	0	1	0
Rythmodan	1	0	0	0	0
Skeletal muscle relaxants	0	0	3	1	0
Sulfonylurea	1	0	0	1	0
Teriflunomide	0	1	0	0	0
Tetrahydrocannabinol	0	0	0	0	1
Unknown	25	29	24	15	21
Vitamins	5	0	0	0	0
Xanthines	0	1	0	0	0

*χ*^2^ = chi-square analysis (215.236); *p* < 0.001 indicates that the differences found were statistically significant.

**Table 7 healthcare-11-01398-t007:** Demographic characteristics of chemical poisoning among patients.

Variables Categories	Frequency	%
Age Group		
1–10	36	41.9%
11–20	41	47.7%
21–30	8	9.3%
≥40	1	1.2%
Gender		
Male	64	74.4%
Female	22	25.6%
Toxic agent		
Clorox	16	18.7%
Clorox + unknown drug	6	6.9%
Dettol + unknown drug	3	3.8%
Alcohol	8	9.3%
Alcohol + unknown drug	1	1.2%
Carbon monoxide	2	2.3%
Chemical material	1	1.2%
Hair dye	5	3.5%
Insecticides + pesticides	15	17.2%
Methanol + kerosine	2	2.4%
Methanol + drug	21	24.4%
Perfume spray	2	2.4%
Mercury	1	1.2%
Paint thinner	1	1.2%
Printer ink	1	1.2%
Unknown	1	1.22%
Type for exposure		
Accidental	81	94.2%
Suicidal	5	5.8%
Region		
Al-Aqiq	4	4.7%
Al-Baha	47	54.7%
Al-Gara	1	1.2%
Al-Hougra	3	3.5%
Al-Mandaq	1	1.2%
Al-Miha	1	1.2%
AL-Mikhwah	4	4.7%
Algara	1	1.2%
Biljurashi	13	15.1%
Namerah	1	1.2%
Qilwa	10	11.6%
Nationality		
Saudi	75	87.21%
Non-Saudi	11	12.8%
Route of Administration		
Dermal	4	4.7%
Inhalation	21	24.4%
Injection	2	2.3%
Swallowing	59	68.6%
Severity of Infection		
Average infection	42	48.8%
Mild infection	16	18.6%
Serious infection	28	32.5%
Recovery rate		
Recovered	85	98.8%
Death	1	1.2%
Antidotes		
Activated charcoal	18	21%
Antiseptic cream	3	3.5%
Atropine	2	2.4%
Hydrocortisone + D5w	4	5.9%
Clexane	10	11.7%
D50w + sodium bicarbonate	8	10.8%
Levoleucovorin calcium	2	15.1%
Methylprednisolone	12	13.2%
N-Acetylcysteine	5	5.7%
No antidote	1	1.2%
O_2_ mask	4	4.7%
Omeprazole	8	9.3%
Potassium chloride in NS	2	2.4%
Pyridoxine hydrochloride B6	2	2.3%
Vitamin K1	5	5.8%
Year of Poisoning		
2019	15	17.4%
2020	37	43.0%
2021	15	17.4%
2022	19	22.1%

**Table 8 healthcare-11-01398-t008:** Correlation between type of chemical toxicant, severity of infection, and antidotes administrated according to gender.

Chemical Toxicant	M	F	χ*^2^*	*p* Value
Clorox	7	9		
Clorox + unknown drug	5	1	46.190	**0.05 ^s^**
Dettol + unknown drug	2	1		
Alcohol	8	0		
Alcohol + unknown drug	0	1		
Carbon monoxide	2	0		
Chemical material	1	0		
Hair dye	3	2		
Insecticides + pesticides	5	10		
Methanol + kerosine	0	2		
Methanol + drug	14	7		
Perfume spray	1	1		
Mercury	1	0		
Paint thinner	1	0		
Printer ink	1	0		
Unknown	1	0		
**Toxicant source**				
Accidental	63	18		
Suicidal	1	4		
**Route of Administration**			8.258	**0.01 ^s^**
Dermal	3	1		
Inhalation	11	10		
Injection	1	1		**0.05 ^s^**
Swallowing	49	10		
**Types of Antidotes**			8.294	
Activated charcoal	9	9		
Antiseptic cream	3	0		
Atropine	2	1		
Hydrocortisone + D5w	3	1		
Clexane	4	3		
D50w + sodium bicarbonate	7	3		
Levoleucovorin calcium	1	1		
Methylprednisolone	11	1		
N-Acetylcysteine	4	1		**0.001 ^s^**
No antidote	0	1		
O_2_ mask	5	0		
Omeprazole	7	1	45.947	
Potassium chloride in NS	2	0		
Potassium hydrochloride B6	3	0		
Vitamin K1	4	1		

M = male; F = female; *χ*^2^ = chi-square analysis; s = significant difference; ns = non-significant difference; significance of *p* values is in bold.

**Table 9 healthcare-11-01398-t009:** Type of exposure of all the cases by age.

Recovery Rate	1–10 y	11–20 y	21–30 y	≥40 y	*χ* ^2^
Average infection	26	6	9	1	28.038 **0.001 ^s^ (*p* value)**
Mild infection	11	4	1	0	
Serious infection	5	12	10	1	
Toxic agent					
Chlorox	8	8	0	0	82.171 **0.01 ^s^ (*p* value)**
Chlorox + unknown drug	1	2	2	1	
Dettol + unknown drug	2	1	0	0	
Alcohol	0	6	2	0	
Alcohol + unknown drug	0	1	0	0	
Carbon monoxide	0	2	0	0	
Chemical material	1	0	0	0	
Hair dye	0	0	0	5	
Insecticides + pesticides	5	5	1	4	
Methanol + kerosine	1	0	1	0	
Methanol + drug	11	4	3	3	
Perfume spray	1	0	1	0	
Mercury	0	0	0	1	
Paint thinner	0	1	1	0	
Printer ink	0	1	0	0	
Unknown	1	0	0	0	
Type for exposure					2.32 0.06 ^ns^ (*p* value)
Accidental	35	37	8	1	
Suicidal	1	4	0	0	
Route of Administration					
Dermal	3	0	1	0	3.632 0.06 ^ns^ (*p* value)
Inhalation	3	11	3	2	
Injection	0	1	0	1	
Swallowing	26	16	16	1	
Type of antidotes					
Activated charcoal	9	7	1	1	41.24 0.942 ^ns^ (*p* value)
Antiseptic cream	2	1	0	0	
Atropine	1	0	1	0	
Hydrocortisone + D5w	2	1	1	0	
Clexane	3	4	0	3	
D50w + sodium bicarbonate	6	1	0	1	
Levoleucovorin calcium	1	0	1	0	
Methylprednisolone	12	0	0	0	
N-Acetylcysteine	0	1	2	2	
No antidote	1	0	0	0	
O_2_ mask	2	1	1	0	
Omeprazole	1	1	4	2	
Potassium chloride in NS	2	0	0	0	
Pyridoxine hydrochloride B6	1	0	1	0	
Vitamin K1	0	1	4	0	

*χ*^2^ = chi-square analysis; s = significant difference; ns = non-significant difference; significance of *p* values is in bold.

## Data Availability

Not applicable.

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
