# Peer review of "A Retrospective Study of Epidemiological Correlations of Food, Drug and Chemical Poisoning in Al-Baha, Western Saudi Arabia"

_healthcare, 2023, doi:10.3390/healthcare11101398_

Round 1

Reviewer 1 Report

Please address the below concerns:

1.       line 50, the author need to have an introductory sentence on food poisoning before “acute contamination occurs….”

2.       line 144, results: total cases reported to be 159, however, 85 male and 75 female cases are reported!

3.       for the place of food poisoning perhaps restaurants and weddings can be considered one category.

4.       In line 159, the authors stated that the “E. coli infection is more prevalent in Al-Baha”, that should correlate with the consumption of contaminated food however the authors continued with:  “implying that people do not consume more contaminated and raw food”.  Please clarify.

5.       Line 201, “This means Bacterial contamination was seen maximum in children having an age in between 1 till 10 years old”. Since all the pathogens reported are bacteria, remove the “bacterial” from this sentence.

6.       The manuscripts needs proofreading for the English language

7.       lines 206-216, the authors categorized the patients based on age range with having severe, average, or mild infection, without mentioning their criteria for having these three groups. Was it based on symptoms? Please clarify and include your criteria in the text.

8.       In lines 220-239, consider having a graph or table to summarize these data.

9.       In line 246, the male-to-female ratio should be 1.1:1, please revise.

10.   line 281, “Drug poisoning was more common in both men and women aged 10–30 years”. Remove

 “more” here, since it was common in both groups.

11. what is the value that is represented by y-axis in figure 3? what are these series 1-4? please explain in details and clarify in the text! Series 4 does not exist in the graph!

Author Response

Regards

Dr Saba

Reviewer 2 Report

The manuscript in question presents serious health risk of poisoning that requires attention to help improve the quality of life in Al-Baha region of KSA. This manuscript is written in good flow but scientifically unjustified to be a scientific publication due to lack of gravity shown by authors where discontinued and very incoherent stats are presented with big differences which might reshape the manuscript at large.  I appreciate the idea of choosing the topic but only taking data and and classifying it can be done very easily. I see no scientific contribution of this work in this presentation. I also see no point in presenting figure 1, it can only be text. This study also did not talk about informed consent and the patient record data it also not attached the diagnosis reports and much of the information is still undercover. Dealing with the data of human subjects also needs ethical approvals and which have also not been explained. Still my decision is to give authors a chance to heavily revise this manuscript addressing all the comments and i am sure this will improve the quality of manuscript.

Abstract

Line 13: "Poisoning has become a common and severe problem across the world" here the nature of poison should be addressed.

Line 18: "rela-tionship" should be reformatted to relationship

Line 29: "Children are still at a significant risk of drug and chemical toxicity." This is an introductory sentence that needs to be replaced.

Introduction

Citations expressed are not validated and inline with respect to appearance. For example [7] appears in a sentence and again it is expressed in the proceeding sentence. I think it should appear once at the end only. Please check the whole draft for these mistakes.

Materials and Methods

As mentioned in the abstract 622 cases were analyzed for frequency of poisoning cases and classisfication but here in line 103 it is state, "with nearly 625 cases counted in total" This might change the over all appearance of data presented in %. Please specify and clarify the data project and be coherent.

Line 144-147: It is stated that you had 159 cases but when you go for the male and female partition you reported 85 and 75 which makes it 160 and also one of it is in bracket and other one is not. The sentence structure and data presentation is also poor.

Table 1: Regional distribution makes it a total of 151 for food poisoned individuals but you claim 159. Stats are totally ignored and non scientifically analyzed. Check this with all similar tables. Moreover, The whole draft needs revision in a manner that number, sums and percentages should be consistently presented.

Author Response

Regards

Dr Saba
